# Effects of Circadian Rhythm and Feeding Modes on Rumen Fermentation and Microorganisms in Hu Sheep

**DOI:** 10.3390/microorganisms10122308

**Published:** 2022-11-22

**Authors:** Chuntao Zhang, Tao Ma, Yan Tu, Shulin Ma, Qiyu Diao

**Affiliations:** 1Key Laboratory of Feed Biotechnology of Ministry of Agriculture and Rural Affairs, Feed Research Institute of Chinese Academy of Agricultural Sciences, Beijing 100081, China; 2Institute of Grain and Oil Crops, Hebei Academy of Agriculture and Forestry Sciences, Shijiazhuang High-Tech Development Zone, Shijiazhuang 050091, China

**Keywords:** circadian rhythm, feeding mode, digestive enzyme activities, rumen microorganisms, Hu sheep

## Abstract

All organisms have a biological clock system which is strongly tied to how well an organism digests food and develops. This study aimed to understand the effects of circadian rhythm and feeding modes on rumen fermentation and microorganisms in Hu sheep. Forty-five healthy Hu sheep were randomly divided into three treatment groups of 15 sheep in each group, wherein they were fed the same concentrate and roughage. Under the condition that the nutrient-feeding amount was consistent throughout the day, the concentrate-to-forage ratio was dynamically adjusted during the day and night. Rumen fluid collected after the feeding experiment was used to determine the study parameters; the results showed a connection between rumen fermentation and the circadian clock. Volatile fatty acids (VFAs), pH, and NH_3_-N were significantly influenced by the fermentation duration (*p* < 0.05). The activities of digestive enzymes also showed a relationship with nutrition and circadian rhythm, and there were differences in the digestive enzyme activities of amylase, lipase, and cellulase (*p* < 0.05). Dominant microorganisms, such as *Saccharomycetes* and *Mucor*, were more abundant in the daytime of the high-concentrate fed group. The correlation among the study objectives was evident from the differences in enzyme activity and microbial diversity among the treatment groups. On the basis of the circadian rhythm characteristics of Hu sheep, changes in the feeding mode of Hu sheep and only adjusting the proportion of concentrate and forage in the morning and evening showed that feeding diets with the high-concentrate ratio in the day significantly reduced rumen PH and increased NH_3_-N concentration (*p* < 0.05). Under this feeding pattern, the activities of major digestive enzymes in the rumen, such as amylase and lipase, were significantly increased (*p* < 0.05), and the microbial diversity was also improved.

## 1. Introduction

Circadian rhythms are involved in biological processes, nutrition, metabolism, and behavioral actions. The law and internal mechanisms known as the biological clock are what make the rhythmic dynamic variations in living events visible [1]. The circadian rhythm is consistent with the earth’s 24-h rotation cycle [2] and crucial to understanding the molecular control, nutrient absorption and metabolism, immunological response, endocrine function, and growth and development of organisms. The daily rhythms of lactation, metabolism and rumen fermentation in dairy cows were significantly affected by different feeding modes [3,4]; time-limited feeding of cows at night altered the circadian rhythms of blood hormones, milk synthesis, and metabolites. It also regulated the stages of milk synthesis by changing the timing of nutrient digestion and absorption, suggesting that molecular clocks in the mammary glands could be reset [5].

China is the largest sheep-raising country in the world, yet there is little research on circadian rhythms in the field of sheep breeding and even fewer on the circadian rhythms of Hu sheep metabolism and rumen environment along the time axis. Currently, meat sheep breeders often ignore the changes occurring in the organs and tissues of animals due to changes in nutrient metabolism at different time periods. It has been reported that diets with different nutrient concentrations, fed in the morning and night, can reduce the emission of fecal and urinary nitrogen, increase the deposited nitrogen, and correspondingly reduce the feed cost accordingly, thus significantly improving the organism’s growth [6]. Therefore, feeding the same standard diet every morning and evening is not always scientific and may result in extra protein and energy consumption due to insufficient absorption, utilization, and conversion of nutrients.

The rumen, the largest digestive organ in ruminants, has a 24 h circadian rhythm and influences the metabolism and breakdown of different feed elements [3]. Rumen fermentation and microbiota are also affected by diet composition and feeding patterns under the influence of diurnal regularity, but there are few reports on this aspect. Therefore in this study, based on the natural habits of ruminants, in this study, we aimed to adjust the whole day’s diet and feeding pattern dynamically. We aimed to provide a reference for fully utilizing Hu sheep and producing high-quality mutton through efficient and effective feeding.

## 2. Materials and Methods

The Chinese Academy of Agricultural Sciences Animal Ethics Committee (permission number: IFR-CAAS20200817) in Beijing, China, supported all experimental plans and procedures. The academy’s standards for using animals in research were strictly followed.

### 2.1. Animals Trial Design and Sample Collection

The experimental Hu sheep were purchased in Huzhou, China. This study was carried out at the Chinese Academy of Agricultural Sciences’ Nankou Pilot Base in Beijing, China (40°21′19.7″ N, 116°45′53.3″ E; 49–74 m height) from September to November 2020, lasting for 70 days. The pretest period lasted for 10 days and the formal period lasted for 60 days. The area has a flat topography and four distinct seasons. In order to accurately perceive the influence of day and night, the experimental sheep were fed outdoors in a single pen on a concrete floor, which prevented the Hu sheep from eating anything other than diet, thus making the experiment more accurate.

Forty-five healthy male Hu sheep (weight 21.57 ± 0.77 kg, 2 months of age) were randomly divided into three treatment groups of 15 sheep each. They were fed the same concentrate and roughage in the ratio of 56:44 (see Table 1 for nutrition level). The sheep were fed twice daily following three different treatments or feeding modes: the control group (CON) Hu sheep were fed 50% of the total concentrate and roughage at sunrise and the remaining 50% at sunset; the DH group was fed 70% of the total concentrate and 30% of the total roughage at sunrise and the rest at sunset, and the DL group was fed 30% of the total concentrate and 70% of the total roughage at sunrise and the rest at sunset. They were provided with clean, fresh water. The study duration was two months, as shown in Table 2. They were kept outdoors to study the influence of the natural diurnal cycle of day and night more effectively.

### 2.2. Sample Collection and Measurements

After the formal test, the rumen juice was collected from the oral cavity using a rumen catheter (ANSCITECH, Wuhan Collebo Animal Husbandry Technology Co., Ltd., Wuhan, China) two hours each before the morning and evening feed. The pH was measured by a portable pH meter (testo-206-pH2.Germany) immediately after filtration through four layers of gauze. To assess rumen fermentation parameters and rumen microbial flora, the remaining samples were transferred to 15 mL and 2 mL centrifuge tubes, respectively, and stored in −20 °C and −80 °C refrigerators at standby.

#### 2.2.1. Rumen Fermentation Parameters

The concentration of NH_3_-N was determined by phenol sodium hypochlorite colorimetry [4]; the concentration of crude microbial protein (MCP) was determined by the purine method [5]; the concentration of volatile fatty acids (VFAs) was determined by gas chromatography [7].

#### 2.2.2. DNA Extraction and 16S rRNA Sequencing

Total bacterial genomic DNA samples were extracted using the Fast DNA SPIN extraction kits (MP Biomedicals, Santa Ana, CA, USA), following the manufacturer’s instructions, and stored at −20 °C before further analysis. The quantity and quality of the extracted DNAs were measured using a NanoDrop ND-1000 spectrophotometer (Thermo Fisher Scientific, Waltham, MA, USA) and agarose gel electrophoresis, respectively.

PCR amplification of the bacterial 16S rRNA V4–V5 region was performed using the forward primer 515F (5′-GTGCCAGCMGCCGCGGTAA-3′) and the reverse primer 907R (5′-CCGTCAATTCMTTTRAGTTT-3′). Sample-specific 7-bp barcodes were incorporated into the primers for multiplex sequencing. The PCR components contained 5 μL of Q5 reaction buffer (5×), 5 μL of Q5 High-Fidelity GC buffer (5×), 0.25 μL of Q5 High-Fidelity DNA Polymerase (5 U/μL), 2 μL (2.5 mM) of dNTPs, 1 μL (10 uM) each of Forward and Reverse primers, 2 μL of DNA Template, and 8.75 μL of ddH2O. Thermal cycling consisted of initial denaturation at 98 °C for 2 min, followed by 25 cycles of denaturation at 98 °C for 15 s, annealing at 55 °C for 30 s, and extension at 72 °C for 30 s, with a final extension of 5 min at 72 °C. PCR amplicons were purified using Agencourt AMPure Beads (Beckman Coulter, Indianapolis, IN, USA) and quantified using the Pico Green dsDNA Assay Kit (Invitrogen, Carlsbad, CA, USA). After the individual quantification step, amplicons were pooled in equal amounts, and paired-end 2 × 300 bp sequencing was performed using the Illumina MiSeq platform with MiSeq Reagent Kit v3. At Shanghai Personal Biotechnology Co., Ltd. (Shanghai, China).

#### 2.2.3. Sequence Analysis

The Quantitative Insights into Microbial Ecology (QIIME, v1.8.0) pipeline was used to process the sequencing data, as previously described [6]. Briefly, raw sequencing reads with exact matches to the barcodes were assigned to the respective samples and identified as valid sequences. Low-quality sequences were filtered using the following criteria: sequences that had a length of <150 bp, sequences that had average phred scores of <20, sequences that contained ambiguous bases, and sequences that contained mononucleotide repeats of >8 bp. Paired-end reads were assembled using FLASH [8]. After chimera detection, the remaining high-quality sequences were clustered into operational taxonomic units (OTUs) with 97% sequence identity using UCLUST. A representative sequence was selected from each OTU using the default parameters. OTU taxonomic classification was conducted by BLAST, searching the representative sequence set against the Greengenes Database using the best hit. An OTU table was further generated to record the abundance and taxonomy of each OTU in each sample. OTUs containing less than 0.001% of the total sequences across all samples were discarded. To minimize the differences in sequencing depth across samples, an average, rounded rarefied OTU table was generated by averaging 100 evenly resampled OTU subsets at 90% of the minimum sequencing depth for further analysis.

### 2.3. Microbial Data Analysis

Sequence data analyses were mainly performed using the QIIME and R packages (v3.2.0). OTU-level alpha diversity indices, such as the Chao1 richness estimator, ACE metric (abundance-based coverage estimator), Shannon diversity index, and Simpson index, were calculated using the OTU table in QIIME. OTU-level ranked abundance curves were generated to compare the richness and evenness of OTUs among the samples. Beta diversity analysis was performed to investigate the structural variation of microbial communities across samples using UniFrac distance metrics [8,9] and visualized via principal coordinate analysis (PCoA), nonmetric multidimensional scaling (NMDS), and the unweighted pair-group method with arithmetic means (UPGMA) hierarchical clustering [10]. Differences in the UniFrac distances for pairwise comparisons among groups were determined using Student’s t-test and the Monte Carlo permutation test with 1000 permutations and visualized through the box-and-whiskers plots. Principal component analysis (PCA) was also conducted based on the genus-level compositional profiles [11]. The significance of differentiation of microbiota structure among groups was evaluated by PERMANOVA (Permutational multivariate analysis of variance) [11] and ANOSIM (Analysis of similarities) [7,11,12] using the R package “vegan”. The taxonomic compositions and abundances were visualized using MEGAN [13] and Graphlan [14]. A Venn diagram was generated to visualize the shared and unique OTUs among samples or groups using the R package “Venn Diagram” based on the occurrence of OTUs across samples/groups regardless of their relative abundance. Taxa abundances at the phylum, class, order, family, genus, and species levels were statistically compared among samples or groups using Meta stats [15] and visualized as violin plots. Linear discriminant analysis effect size (LEfSe) was used to detect differentially abundant taxa across groups using the default parameters [16]. Partial least squares discriminant analysis (PLS-DA) was also introduced as a supervised model to reveal the microbiota variation among groups, using the “plsda” function in the R package “mix Omics”. Random forest analysis was applied to discriminate the samples from different groups using the R package “random forest” with 1000 trees and all default settings [17]. The generalization error was estimated using 10-fold cross-validation. The expected “baseline” error, which was obtained using a classifier that simply predicts the most common category label, was also included. Co-occurrence analysis was performed by calculating Spearman’s rank correlations between the predominant taxa. Correlations with |RHO| > 0.6 and *p <* 0.01 were visualized as co-occurrence networks using Cytoscape [18]. Microbial functions were predicted by phylogenetic investigation of communities by reconstruction of unobserved states (PICRUST based on high-quality sequences [19].

## 3. Results

### 3.1. Rumen Fermentation Parameters

The rumen pH of all the experimental sheep was higher during the day than that at night; among them, the pH of the DH group, which was fed with 70% of the concentrate in the morning, was higher than that observed for the other two treatment groups, DL and CON. The DH group also reported significantly higher values of NH_3_-N than that observed in the other two groups during the day. However, the values were highest for the CON group at night. The MCP values for all the treatment groups were higher during the day than those at night; The daytime yield and the total yield for day and night were both highest for the DH group. The results indicated that the rumen fermentation was affected by both day and night and feeding mode, and the interaction between the two has a significant effect on pH and NH_3_-N (Table 3).

The Circadian rhythm and feeding mode had significant effects on the total VFA (Table 4). The influence of circadian rhythm on the rumen acetic acid, propionic acid, isovaleric acid, total VFA, and the ratio of acetate to propionic acid was significantly higher during the day than at night (*p* < 0.05). Effects of feeding method: the contents of total volatile fatty acids, acetic acid, and propionic acid in rumen fluid in the DH group were significantly higher than those in CON and DL groups (*p* < 0.05) and lower than the other two groups at night (*p* > 0.05). The interaction of circadian rhythm and feeding mode had significant effects on acetic acid, propionic acid and total VFA (*p* < 0.05).

### 3.2. Rumen Digestive Enzyme Activity

The effect of the various meal schedules on ruminal digestive enzyme activity is depicted in Table 5. The activities of amylase, lipase, and cellulase were considerably greater during the day than those during the night (*p* < 0.05), such that amylase was 44.63 (U/g) vs. 33.47 (U/g), lipase was 64.43 (U/g) vs. 57.36(U/g), and cellulase was 44.63 (U/g) vs. 2.65 (U/g) each for day vs. night, respectively.

Different feeding modes led to different results: the DH group, fed a greater percentage of the concentrated feed in the morning, showed a 24.84% higher daytime amylase activity than that of the CON group and 28.99% higher than that of the DL group (*p* < 0.05). The DL group, fed a high percentage of the concentrate at night, showed higher lipase and cellulose levels than those of the other two treatment groups, both at night and throughout the day. The interaction between circadian rhythm and feeding pattern had the most significant effect on amylase (*p* < 0.05).

### 3.3. Rumen Microflora

The results from 16S, ITS, 18S rRNA high-throughput sequencing, and 18S sequencing analysis revealed that rumen microbiota showed diurnal variation characteristics. The Shannon index of rumen bacteria, fungi, and protozoa was higher during the day than at night (Figure 1). For OTU values, bacteria showed higher values during the day than at night, while fungi and protozoa showed the opposite trend (Figure 2). Regarding beta diversity, no significant changes in community membership and structure were observed at unweighted UniFrac distance based on principal coordinate analysis (PCoA) (Figure 3). At the phyla level: Bacteroidetes and Mucoromycota were higher during the day than at night, while Firmicutes were higher at night than during the day. At the genus level: *Prevotella_1*, *Mucor*, and *Entodinium* were higher during the day than at night, while *Rikenellaceae_RC9_gut*, *Ruminococcaceae_NK4A214*, and *Epidinium* were higher at night than during the day (Figure 4).

The feeding mode affected rumen microorganisms. A further indication that there was no significant variation in the makeup of the bacterial community in the rumen habitat was provided by the PCoA data that revealed that the distance between the day and night samples was not entirely separated (*p* > 0.05). The OTU value of DL was high both day and night (Figure 2). At the phylum level, *Bacteroidetes* and *Mucoromycota* abundance in the DH group increased and was greater in the morning than that at night. However, *Firmicutes* and *Basidiomycota* abundances declined and were lower than those in the other two groups. Compared to the other two groups, the DH group had a considerably higher abundance of *Prevotella_1*, *Saccharomyces*, and *Entodinium*, while the abundance of *Ruminococcaceae_NK4A214* and *Epidinium* species was observed to decrease.

## 4. Discussion

The structure of the rumen microflora affects the metabolic homeostasis of the body along with the diurnal fluctuations, even though the microorganisms are not directly exposed to the diurnal light and dark environment. The mechanisms by which rhythms in the microbiota affect the diurnal activity of the host are still unknown. In this study, the interaction between the host and the rumen environment was investigated, which assisted in finding the primary rumen microbiota during the day and at night and which rumen microbes are predominant after a change in the feeding mode.

### 4.1. Circadian Rhythm Presented Effects on Rumen Environment, Enzyme Activity, and Rumen Microflora

#### 4.1.1. Circadian Rhythm Presented Effects on Rumen Environment

In this study, day and night showed an effect on the pH and NH_3_−N concentrations, which were both within the usual range. The results of Sato et al. [20], who employed a wireless data pH monitoring device to constantly assess the circadian rhythm of rumen pH in dairy cows, were similar to the findings in this study that rumen pH is higher during the day than at night, regardless of the variations in feeding habits. The shift in NH_3_-N was logically the opposite of pH and MCP, but the diurnal variation in MCP was compatible with pH, which was greater during the day than that at night.

Gustafsson [21] determined that the NH_3_-N concentration peaked between 6:00 am and 8:00 am after measuring rumen fermentation parameters in three cows. His research also showed that feeding schedules and meal makeup had an impact on the diurnal changes in NH_3_-N concentration. As a result of high metabolism, high digestive enzyme activity, and high digestibility and absorbability of the feed, a circadian property of volatile fatty acids, the concentrations of propionic and acetic acid were higher during the day than that at night.

#### 4.1.2. Circadian Rhythm Presented Effects on Enzyme Activity

Enzymes secreted by microorganisms are manifestations of their physiological activities [22]. The various enzymes secreted by microorganisms participate in the ruminal digestion of nutrients present in the feed [23,24]. In this study, we found that amylase and lipase activities were higher during the day than those at night, indicating that the day and night, light and dark environments had a direct influence on the activity of rumen digestive enzymes. During the day, rumen microorganisms are more active and need to quickly obtain more energy to form feedback regulation and promote rapid degradation of relevant digestive enzyme substrates to provide energy and precursors. The degradation process of fibers in the rumen is extremely complex, and ruminants use various anaerobic microorganisms and digestive enzymes to decompose crude fiber to obtain energy [25]. Additionally, we found that cellulase activity in the rumen is high during the day, and it may be attributed to the strong behavioral activity and metabolism that sheep have during the day.

#### 4.1.3. Effects of Circadian Rhythm on Rumen Microflora

As early as 1987, some scholars reported that the emptying speed of the same individual after eating at 8:00 pm was significantly lower than that after 8:00 am [26]. Lindberg et al. [27] studied the dynamic electrogastrogram of 30 volunteers and found significant diurnal changes, indicating that the average frequency at noon was significantly higher than that at night, suggestive of stronger rumen functional activity during the day and a correspondingly enhanced metabolism. Recently, a large number of studies were carried out on the characteristics of diurnal variations in gastrointestinal microbiota, mainly focused on monogastric mammals, such as mice and humans. However, studies on the microbiota of ruminants are still lacking.

Sequence analysis revealed that the predominant rumen bacteria were *Bacteroides* and *Firmicutes* and that their abundance had specific circadian features. When we looked at the phylum level, we discovered that *Bacteroidetes* and *Mucoromycota* were more abundant during the day than at night; *Firmicutes* were more abundant at night than during the day; and *Prevotella_1*, *Mucor*, and *Entodinium* were more abundant during the day. Furthermore, *Rikenellaceae_RC9_gut*, *Ruminococcaceae_NK4A214*, and *Epidinium* were higher at night. As all organ tissues of the body are in an exciting working state during the day, the number of rumen fluid protozoa decreased with the increase in feed intake of Hu sheep during the day. However, pectinase and cellulase activities increased slowly, while amylase activity increased significantly [24]. Liang et al. [28]. Quantitative tests revealed that the diurnal variation of *Bacteroides* was the main driving force of diurnal variation in microbiota composition, which was further confirmed by the number and composition of diurnal variations in microbiota.

### 4.2. Effects of Feeding Mode (Experimental Treatment) on Rumen Internal Environment, Enzyme Activity, and Rumen Microflora Based on Circadian Rhythm

#### 4.2.1. Effects of Feeding Mode on Rumen Environment Based on Circadian Rhythm

Numerous studies have demonstrated that the internal environment and associated circadian rhythms in animal rumen vary when external factors (such as feeding time, feed shape and composition, and temperature) are altered [28,29]. Our research revealed that the rumen pH of sheep in the DH group was significantly lower than that of the other two groups, both during the day and at night. This was because the higher level of concentrate and the lower neutral detergent fiber content fed at sunrise could shorten the chewing time of the ruminants, accelerate the rate of feed degradation, increase the concentration of VFA and propionic acid, and reduce the rumen pH with the flow of feed in the stomach. The results were exactly the opposite for DL: the rumen pH was higher at sunset; when a high amount of concentrate was fed to this group, the pH dropped significantly, indicating that the difference in CP gradient may be the cause of the diurnal variation in rumen pH. Our finding conflicts with the study by Dasilcal et al., which indicates that dietary CP levels have no impact on rumen pH [30].

Khafipour et al. [31] found that a high proportion of concentrate in the feed can significantly upregulate the expression of biological clock genes (BMAL1, PER1, CLOCK, and tRNA), change rumen acidity, and affect the absorption of VFA. These findings suggest that feeding mode and circadian rhythm have some bearing on volatile acids. In this study, acetic acid was found to have a positive correlation with the amount of roughage. When the acetic acid concentrations were compared, the DL group had a greater acetic acid concentration than the other two groups. The concentration of propionic acid demonstrated a favorable correlation: when the proportion of concentrate fed at night was low, the propionic acid level was the lowest. In contrast, when the proportion of concentrate fed at sunrise was high, the propionic acid content was noticeably greater than that of the other two groups. Total volatile fatty acids did not significantly differ among the three groups. However, further experimental support is required to elucidate the regulation of the VFA concentration mechanism.

#### 4.2.2. Effects of Feeding Mode on Enzyme Activity Based on Circadian Rhythm

Following Li Yuqi et al. [32], in sheep, feeding causes rumen protease and cellulase activities to first rise and subsequently fall, showing that food has a direct impact on rumen digestive enzyme activity. The amount of concentrate in the diet affected amylase activity. At dawn, feeding 70% concentrate and 30% roughage promoted amylase secretion and activity. Amylase had the highest enzyme activity, suggesting that altering the feeding style would likewise affect rumen enzyme activity. Following Wang Xiaojuan’s research [33], which is consistent with the findings of this experiment, feeding groups with a fine-to-coarse ratio of 70:30 exhibited much greater amylase activity than that observed in the 30:70 group. The shortcoming of this study was the absence of a 24 h comparison of the activity of rumen enzymes. At dawn, when cellulase activity was at its peak, 30% of refined feed and 70% of coarse feed were supplied. Additionally, the sheep may only require more chewing and ruminating and the increased fragmentation of the meal would increase cellulase secretion and degradation [34]. The findings indicated that several variables, including the ratio of concentrate to roughage and feeding duration, might have an impact on how the primary digestive enzyme activity in the rumen changes over time.

#### 4.2.3. Effects of Feeding Mode on Rumen Microflora Based on Circadian Rhythm

The rumen microbiome changes with alterations in the feeding mode. The PCoA results revealed that the distance between the day and night samples was not separated, further indicating that the bacterial community composition in rumen habitats did not differ significantly between the two (Figure 2); however, the OTU value for higher coarse material ratio (DL) was high for both day and night values. At the phylum level, the DH group, fed with a high concentrate in the morning, saw a rise in *Bacteroidetes* and *Mucoromycota* abundance and was greater than that at night, while *Firmicutes* and *Basidiomycota* abundance declined and was lower than that in the other two groups. At the genus level, the DH group, fed with a high concentrate in the morning, had a considerably greater abundance of *Prevotella_1*, *Saccharomyces*, and *Entodinium* than did the other two groups. Compared to the other two groups, the abundance of *Ruminococcaceae_NK4A214* and *Epidinium* species decreased.

We also found that *Bacteroideas* and *Firmicutes* are the two biggest phyla independent of the change in feeding method, and they also make up the majority of the gut microbiota in ruminants of various ages [35], a stat which is correlated with the amount of body fat. By lowering the number of bacteria and raising the content of *Firmicutes* in the diet, which is thought to be the primary factor impacting the microbiota, the prevalence of obesity can be reduced [36,37]. Numerous plant oligosaccharides and polysaccharides, host-derived glycans, and oligosaccharides from breast milk may all be fermented by *Bacteroides*. A diet rich in protein, sugar, and carbohydrates is associated with a high abundance of *Bacteroides* [38]. With a high proportion of concentrate at sunrise in the diet, the abundance of *Bacteroidetes* increased and was higher than that at night, while the abundance of *Firmicutes* was lower than that in the control group, showing that the amount of roughage and diurnal variation had an impact on the abundance of flora. These test results, however, cannot be compared with those of the other studies since there is still a gap in our understanding of the molecular mechanisms governing the rumen flora at different times of the day (and night) and their effect on the real output. Starch and plant cell wall polysaccharides like xylan and pectin can be broken down and used by *Prevotella_1* in the rumen. When the feeding mode is switched, and a high percentage of concentrate is fed in the morning, *Prevotella* abundance rises, and the three groups’ floral abundance is higher than that at night. The primary fiber-degrading bacteria in the rumen is *Ruminococcaceae_NK4A214,* and a daily diet high in forage feed can enhance the relative abundance of this bacterium. The intestinal tract is home to the symbiotic bacterium *Rikenellaceae_RC9_gut*; its presence, even in very modest amounts, suggests a healthy gastrointestinal system. The group with the highest concentration of concentrate fed at sunrise, the DH group, had the lowest concentration of *Rikenellaceae_RC9_gut*. The abundance of *Rikenellaceae_RC9_gut* was significantly higher at sunset in the DL group fed with a higher proportion of coarse feed in the evening. This suggests that the feeding pattern promoting the growth of *Rikenellaceae_RC9_gut* was more suited to a healthy gastrointestinal state.

The richness of fungal flora in the rumen can be improved by a meal that has a high proportion of concentrate at sunrise. Even at low percentages, fungi can increase the effectiveness of rumen digestion by utilizing and degrading refractory polymer components, such as cellulose, hemicellulose, and lignin, in the feed by creating a range of highly active enzymes [39]. The rate of roughage breakdown and cellulase activity in ruminants’ rumens was considerably reduced when anaerobic fungi were eliminated. There are two major fungi also in this feeding mode: *Saccharomyces* and *Mucor*. Ruminants’ rumen environments can be improved by consuming yeast preparations. *Ochratoxin* may be broken down by *Saccharomyces*, and certain yeasts can also break down fumonisins, zearalenone, and vomiting toxins [40]. Sousa et al. [41] fed active yeast to grazing cattle for a year and the findings demonstrated that the presence of active yeast promoted the development of bacteria that broke down fiber and increased the digestibility of fiber for grazing animals, regardless of the season. According to Thaiss et al. [42], rumen fungal populations are affected significantly by various food patterns. The findings of this experiment were comparable, and the number of rumen fungi in animals fed high-cellulose diets was higher than that in animals fed low-cellulose diets. In addition, we discovered that while the floral composition of protozoa in animals is more straightforward than that of bacteria, it is nevertheless influenced by the time of day and the type of feeding. For instance, the proportion of concentrate fed is positively correlated with the quantity of *Entodinium*, whereas the proportion of roughage fed is positively correlated with the abundance of *Epidinium*. This is because *Entodinium* prefers to degrade the starch while *Epidinium* prefer to degrade the plant cell wall [43]. Rumen ciliate movement is crucial for controlling rumen fermentation because it can increase the activity of the fungal population, maintain the rumen acid-base balance and osmotic pressure, and lessen the impact of acidosis brought on by fast starch fermentation. The rumen microecology evolves as different groups engage with one another under particular circumstances. In order to conduct a comprehensive study that helps understand the essence of microorganisms and is beneficial in the production of ruminants. Furthermore, it is necessary to link the composition and function of rumen microbes with the features of the circadian rhythm.

## 5. Conclusions

In this study, we characterized the effects of circadian rhythms and feeding patterns on the rumen environment. Rumen fermentation parameters, enzyme activity, and microbiota were higher during the day than at night, indicating a relationship between the circadian cycle and the rumen environment. The internal environment of the rumen was also affected by the feeding mode. A high concentration of the concentrated feed in the morning can boost the rumen’s production of total VFA, NH_3_-N, and MCP as well as the key digestive enzymes’ activity and microbial diversity, which provides insights that reference for the precision regulation of nutrition in aquaculture.

## Figures and Tables

**Figure 1 microorganisms-10-02308-f001:**
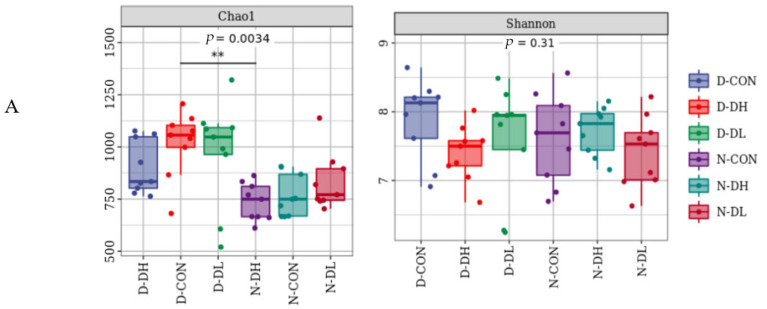
Diversity of rumen microflora in mutton sheep fed with different diets. * Stand for significant (*p* < 0.05), ** stand for extremely significant (*p* < 0.01).Based on the Shannon index and observations from the longitudinal changes in the intestinal microbial community of operational taxon units (OTUs), the meanings of symbols in the box line diagram are as follows: the upper and lower end lines of the box, and the upper and lower quartile range (IQR); Median line, median; Upper and lower edges, maximum and minimum values (extreme values within 1.5 times the IQR range); points outside the upper and lower edges represent outliers. The number under the diversity index label is the *p*-value of the Kruskal–Wallis test. (**A**) represents the alpha diversity index of bacteria, (**B**) represents the alpha diversity index of fungi, and (**C**) represents the alpha diversity index of protozoa. D−DH, D−CON, and D−DL represent the alpha diversity index of daytime of the DH, CON, and DL groups, respectively; and N−DH, N−CON, and N−DL represent the alpha diversity index of the DH, CON and DL groups at night.

**Figure 2 microorganisms-10-02308-f002:**
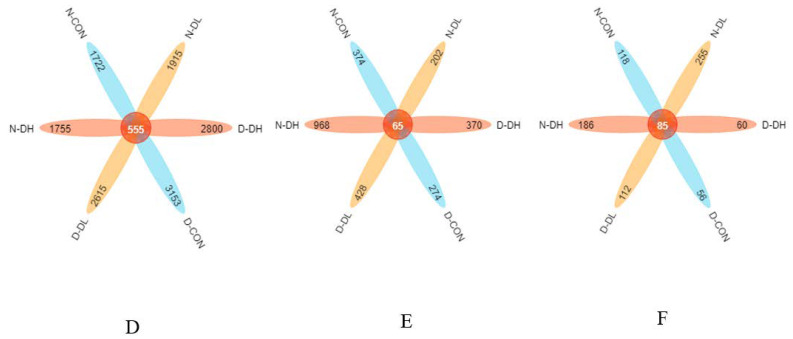
Each color block represents a group. The overlapping area between the color blocks indicates the ASV/OTU shared by the corresponding groups, and the number of each block indicates the number of ASV/OTU contained in the block. In the figure, (**D**) represents bacteria, (**E**) represents fungi and (**F**) represents protozoa; D represents 12 h of the day, and N represents 12 h at night.

**Figure 3 microorganisms-10-02308-f003:**
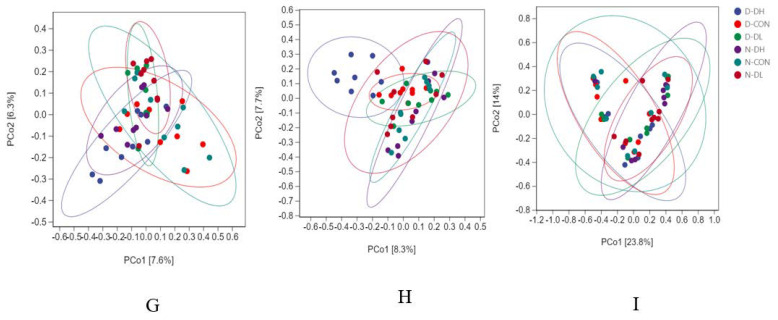
Structure of rumen microflora of mutton sheep fed with different diets. PCoA is used to assess the differences in bacterial community composition between samples and to visualize the potential aggregation of samples. (**G**–**I**) represent the community structure of bacteria, fungi and protozoa, respectively. Each point represents a unique sample. D−DH, D−CON, and D−DL represent 12 h of the and N−DH, N−CON and N−DL represent 12 h of the night (and are distinguished by blue, red, green, and purple colors. Based on the OTU composition and abundance, the unweighted UniFrac distance was used for the calculations. The percentage in the brackets of the coordinate axis represents the proportion of the sample difference data (distance matrix) that can be explained by the corresponding coordinate axis. A projection analysis is suggested (the distance is not to be taken literally): the closer the projection distance of the two points on the coordinate axis, the more similar the community composition of the two samples in the corresponding dimension. On the contrary, the greater the distance between the two points, the lower the similarity between them.

**Figure 4 microorganisms-10-02308-f004:**
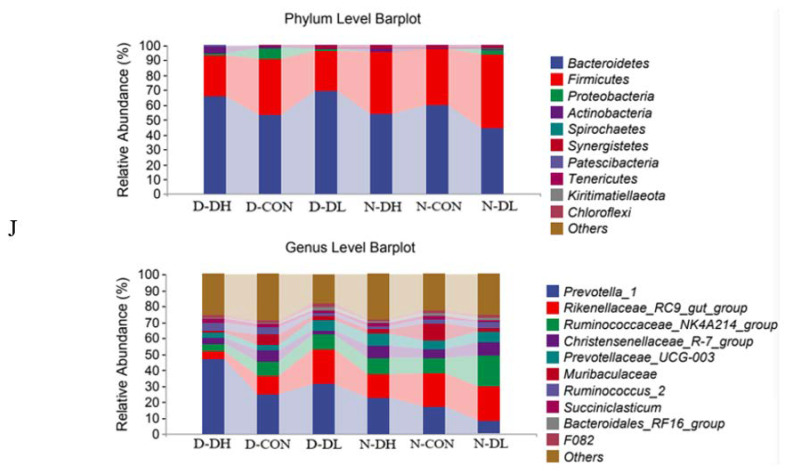
Composition of rumen microflora at phylum and genus level in mutton sheep fed with different diets. In the figure, the abscissa in the figure is the name of each sample/group/whole of the grouping scheme, and the ordinate is the relative abundance of each taxon at a specific classification level. (**J**) represents bacteria, (**K**) represents fungi, and (**L**) represents protozoa; D represents 12 h of the day, and N represents 12 h at night.

**Table 1 microorganisms-10-02308-t001:** Composition and nutrient levels of diets (DM basis%).

Items	Ingredients		
	Concentrate (%)	Leymus Chinensis (%)	Concentrate + Leymus Chinensis (%)
Dry matter, DM/(%)	94.77	90.93	93.08
Organics matter, OM/(%)	88.93	93.26	90.84
Metabolic energy, ME/(MJ/kg)	11.52	5.93	9.06
Crude protein, CP/(%)	19.27	7.60	14.14
Ether extract, EE/(%)	3.43	2.45	3.00
Neutral detergent fiber, NDF/(%)	18.46	65.90	39.33
Acid detergent fiber, ADF/(%)	14.34	40.62	25.90
Crude ash Ash/(%)	11.07	6.74	9.16
Calcium, Ca/(%)	0.81	0.50	0.71
Phosphorus, P/(%)	0.49	0.08	0.31

➀ Concentrate produced by the Graminifeng Company, concentrate mainly consists of corn, soybean meal, expanded soybean, cotton meal, corn DDGS, calcium hydrogen phosphate, stone powder, sodium chloride, lysine, vitamin A, vitamin D3, ferrous sulfate, copper sulfate and so on. Leymus chinensis was purchased from Beijing Sanshi Dairy Farm on an air-dried basis. ➁ The ratio of concentrate to Leymus chinensis was 56:44. ➂ The calculated metabolizable energy and the rest are measured values.

**Table 2 microorganisms-10-02308-t002:** Experimental design of circadian rhythm (Feeding mode).

Groups	Diets	
	Sunrise	Sunset
CON	50% total concentrate + 50% total roughage	50% total concentrate + 50% total roughage
DH	70% total concentrate + 30% total roughage	30% total concentrate + 70% total roughage
DL	30% total concentrate + 70% total roughage	70% total concentrate + 30% total roughage

**Table 3 microorganisms-10-02308-t003:** Correlation among circadian rhythm and rumen pH, NH_3_-N, and MCP in HU Sheep.

Items	Time Period	Groups			SEM	*p*-Value		
		CON	DH	DL		Group	Time	Group ∗ Time
pH	Day	6.84 ^a^	6.26 ^b^	7.02 ^a^	0.12	<0.01	<0.01	0.05
	Night	6.30 ^a^	6.12 ^b^	6.38 ^a^				
NH_3_-N/(mg·dL^−1^)	Day	10.40 ^b^	17.91 ^a^	10.08 ^b^	0.75	<0.01	0.01	<0.01
	Night	11.18	10.56	11.46				
MCP/(mg·mL^−1^)	Day	1.39	1.78	1.42	0.01	0.82	0.12	0.79
	Night	1.09	1.08	1.11				

CON: Hu sheep were fed 50% of the total concentrate and roughage at sunrise and the remaining 50% at sunset; DH: Hu sheep were fed 70% of the total concentrate and 30% of the total roughage at sunrise and the rest at sunset; DL: Hu sheep were fed 30% of the total concentrate and 70% of the total roughage at sunrise and the rest at sunset. In the table, no letters or the same letters in peer data indicate no significant difference (*p* > 0.05), while different lowercase letters indicate significant difference (*p* < 0.05). The following as the same.

**Table 4 microorganisms-10-02308-t004:** Correlation between circadian rhythm and rumen VFA in HU sheep.

Items	Time Period	Groups			SEM	*p*-Value		
		CON	DH	DL		Group	Time	Group ∗ Time
Acetic acid/(%)	Day	58.19	59.54	60.23	0.01	0.84	<0.01	<0.01
	Night	61.17	57.69	56.29				
Propionic acid/(%)	Day	19.06 ^b^	26.76 ^a^	19.06 ^b^	0.23	<0.01	<0.01	<0.01
	Night	17.62	18.08	18.57				
Isobutyric acid/(%)	Day	12.50 ^ab^	12.45 ^a^	9.86 ^b^	0.13	0.05	0.63	0.46
	Night	10.59	16.61	12.49				
Butyric acid/(%)	Day	8.76 ^a^	1.86 ^b^	9.55 ^a^	0.47	<0.01	0.23	0.05
	Night	8.57 ^a^	6.98 ^b^	9.23 ^a^				
Isovaleric acid/(%)	Day	0.83	0.81	0.78	0.03	0.06	<0.01	0.11
	Night	1.34	1.39	1.20				
Valeric acid/(%)	Day	0.66 ^b^	1.06 ^a^	0.52 ^b^	0.15	0.03	0.06	<0.01
	Night	0.74	063	0.63				
TVFA/(mmol·L^−1^)	Day	59.22 ^b^	81.31 ^a^	65.53 ^b^	0.34	0.47	<0.01	<0.01
	Night	65.08	51.16	59.07				
A/P	Day	3.12 ^a^	2.52 ^b^	3.22 ^a^	0.03	0.005	<0.01	0.55
	Night	3.48	3.17	3.53				

**Table 5 microorganisms-10-02308-t005:** Rumen digestive enzyme activity index.

Items	Time	Groups			SEM	*p*-Value		
		CON	DH	DL		Group	Time	Group ∗ Time
Amylase (U/g)	Day	35.75 ^b^	44.63 ^a^	34.60 ^b^	0.32	<0.01	<0.01	<0.01
	Night	31.44 ^c^	33.47 ^b^	36.71 ^a^				
Lipase (U/g)	Day	61.42 ^b^	64.43 ^ab^	66.77 ^a^	0.30	0.06	<0.01	0.05
	Night	59.2	57.36	59.49				
Cellulase (U/g)	Day	4.43	4.44	4.85	0.15	0.06	<0.01	0.42
	Night	3.38 ^ab^	2.65 ^b^	3.66 ^a^				

## Data Availability

The data that support the findings of this study are available on request from the corresponding author. The data are not publicly available due to privacy or ethical restrictions.

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
