# Peer review of "Effects of Circadian Rhythm and Feeding Modes on Rumen Fermentation and Microorganisms in Hu Sheep"

_microorganisms, 2022, doi:10.3390/microorganisms10122308_

Round 1

Reviewer 1 Report

This MS studied the effects of circadian rhythm and feeding modes on rumen fermentation and microorganisms in Hu sheep,which  is a very attractive topic. But some areas need to be improved.

The interaction of feeding method and time was significatly for most indexes, but  it was rarely mentioned in the results and discussions. Please detail it .

L16 "concentrate to crude" should be "concentrate to forage"

L20 TVFA should be total VFA

L23 in the daytime of the high-concentrate fed group

L40 "different feeding time and feeding times" not clear

L175 is it right to use "during the day and at night"?

Table 3-5 please correct the format error, and the meaning of "DH and DL" was not clear, please add the note of table.

L187 "The ci C"  ??

L189 "ethylene to propyl" ??

L200-1  please add the unit

L212 adjusted as "16S, ITS , and 18S rRNA high-throughput sequencing"

L214 the Shannon index were not different for fungi or protozoa in the figure 1, Please check.

L221 Please delete the sentence.

L290  Change as "circadian rhythm presented effects"

L309 how to prove "light and dark environments had a direct influence on the activity of rumen digestive enzymes"

L328-33 this is result not discussion

L450-453, please disscuss the result with the ref. of Li et al. 2022, that the substrate preferences of Entodinium and Epidinium was different, Entodinium prefer to degrade the starch while the Epidinium prefer to degrade the plant cell wall.
Li, Z., X. Wang, Y. Zhang, Z. Yu, T. Zhang, X. Dai, X. Pan, R. Jing, Y. Yan, Y. Liu, S. Gao, F. Li, Y. Huang, J. Tian, J. Yao, X. Xing, T. Shi, J. Ning, B. Yao, H. Huang, and Y. Jiang. 2022. Genomic insights into the phylogeny and biomass-degrading enzymes of rumen ciliates. ISME J 1–13.  doi:10.1038/s41396-022-01306-8.

Author Response

Manuscript ID: microorganisms-2011318

Title: “Effects of circadian rhythm and feeding modes on rumen fermentation and microorganisms in Hu Sheep”.

Response to Reviewer Comments

Dear Editor: 

Great thanks to you and the reviewers providing us nice comments as well as revision suggestions.

We would like to express our appreciation to Reviewer for carefully reviewing our article and providing useful comments and suggestions.

We have made revisions to the revised manuscript in response to the comments and recommendations received. We have copied each reviewer's comment below and followed up with a comprehensive answer Additionally, we have used red colour in the revised portions of the manuscript by using “Track Changes” function.

Hopefully the revised version could be acceptable, and if there are any improper revision, please don’t hesitate to inform us.

Great thanks again,

Qiyu Diao

Key Laboratory of Feed Biotechnology of Ministry of Agriculture and Rural Affairs, Feed Research In-stitute of Chinese Academy of Agricultural Sciences, Beijing, China

Response to Reviewer  Comments

Point 1: The interaction of feeding method and time was significatly for most indexes, but  it was rarely mentioned in the results and discussions. Please detail it.

Response 1: Yes, accepted and revised. the feeding pattern change is based on the circadian rhythm (feeding time), we have modified and detailed accordingly.

Point 2: L16 "concentrate to crude" should be "concentrate to forage"

Response 2: Yes, accepted and revised. Corresponding modifications have been made here and the whole manuscript has also been checked.

Point 3: L20 TVFA should be total VFA

Response 3: Yes, accepted and revised. Corresponding modifications have been made here and the whole manuscript has also been checked.

Point 4: L23 in the daytime of the high-concentrate fed group

Response 4: Yes, accepted and revised, thanks.

Point 5: L40 "different feeding time and feeding times" not clear

Response 5: Yes, accepted and revised “different feeding time and feeding times” to “different feeding mode”. The feeding time was the time of morning and evening feeding, and the feeding times were the number of feeding times in a day.

Point 6: L175 is it right to use "during the day and at night"?

Response 6: Yes, it is right. Our experimental design for twice daily feeding, feeding for the first time at sunrise, the sunset on the second feeding, sunrise to sunset this paragraph of time is called the day, to the second day of the sunrise sunset called night, sampling time before sunrise, represents the fermentation index in the night, and collected before sunset, is representative of the rumen environment during the day, Therefore, Day and night are used in the article. Thanks.

Point 7: Table 3-5 please correct the format error, and the meaning of "DH and DL" was not clear, please add the note of table.

Response 7: Yes, accepted and revised .This part has been added.

Point 8: L187"The ci C"  ??

Response 8: Here is a misspelling of a word, the correct word is circadian, so through you comment, we modified this part. Thanks.

Point 9: L189"ethylene to propyl" ??

Response 9: Yes, accepted and revised ,this part modified. Thank you very much.

Point 10: L200-1  please add the unit

Response 10: Yes, accepted and revised .This part has been added. Thanks!

Point 11: L212 adjusted as "16S, ITS , and 18S rRNA high-throughput sequencing"

Response 11: Yes, accepted and revised .This part modified, thanks.

Point 12: L214 the Shannon index were not different for fungi or protozoa in the figure 1, Please check.

Response 12: Yes, accepted and revised. We have checked this part. Comparing the diurnal changes of bacteria, fungi and protozoa, we chose the control group (eating the same diet in day and night, that is, feeding the same proportion of concentrate and coarse feed in morning and evening). Figure 1 shows that although there is no statistically significant difference between morning and evening, However, the results showed that Shannon index in daytime was higher than that at night. Thanks.

Point 13: L221 Please delete the sentence.

Response 13: Yes. It has been deleted. thanks.

Point 14: L290 Change as "circadian rhythm presented effects"
Response 14: Yes, accepted and revised . It has been changed.

Point 15: L309 how to prove "light and dark environments had a direct influence on the activity of rumen digestive enzymes"

Response 15: Yes, thank you for your question. Our interpretation is that Photoperiodic change is the most obvious and continuous signal factor in the environmental changes brought about by the rotation of the earth. Therefore, light is considered to be the most influential factor on the body. Changing the daytime light time in the circadian rhythm will have a certain impact on the behavior, metabolism, growth and development of domestic animals. [Previous research progress] Studies have shown that, the physiological function of sheep in the morning period is stronger than that in the afternoon period. At this time, the body oxygen supply is sufficient, the heart and liver function is good, the enzyme activity is strong, and the digestion and metabolism are at the best stage. When the rhythm of the rumen biological clock changes and the optimal PH of the rumen is destroyed, the activities of a series of digestive enzymes (amylase, protease, lipase, etc.) and non-digestive enzymes (cellulase, pectinase, etc.) will be affected, and the catalytic effect of enzymes will be inhibited.

Point 16: L328-33 this is result not discussion

Response 16: Yes, accepted and revised. It has been deleted.

Point 17: L450-453, please disscuss the result with the ref. of Li et al. 2022, that the substrate preferences of Entodinium and Epidinium was different, Entodinium prefer to degrade the starch while the Epidinium prefer to degrade the plant cell wall.

Response 17: Yes, accepted and revised, This part modified. Thanks.  

Reviewer 2 Report

Research to determine how arcadian rhythms affect the rumen is an important subject that needs to be researched.  Much more detail needs to be included in the methods section of this paper.  Details on how the feed was prepared and how the sheep were housed can totally change the results. This reviewer has observed, during the review of many papers, that the methods sections often lack important details that can change the results.

Line 71 - Add information on the origin of the Hu sheep and the feeds they were fed before your experiment started.

Line 76 - include approximate types of sunrise and sunset and the time of year the experiment was conducted.

Line 76 - State exactly what the concentrate feed was. If it was a commercial product, state its name and the main ingredients.  The ingredients in the concentrate and the method used to prepare the concentrate will have an effect on your results.

Line 80 - Describe the environment the sheep were housed in during your experiment.  Things the sheep could eat in the environment, such as dirt, could have an effect on rumen microorganisms.

Table 1 - Describe the wild rye. This is a feed this reviewer is not familiar with.  Please describe it.  Is it a dried straw or hay?

Line 87 - Add manufacturer of the rumen catheter.

Line 90 - Add manufacturer and model number of the pH meter.

This reviewer wants to help researchers improve the methods sections of their papers.  Many problems in science, where there has been a failure to replicate the results, has been due  to poor methods sections that do not provide a complete description.  A reviewer who is an expert in rumen microbiology should also review this paper.

Author Response

Manuscript ID: microorganisms-2011318

Title: “Effects of circadian rhythm and feeding modes on rumen fermentation and microorganisms in Hu Sheep”.

Response to Reviewer Comments

Dear Editor: 

Great thanks to you and the reviewers providing us nice comments as well as revision suggestions.

We would like to express our appreciation to Reviewer for carefully reviewing our article and providing useful comments and suggestions.

We have made revisions to the revised manuscript in response to the comments and recommendations received. We have copied each reviewer's comment below and followed up with a comprehensive answer Additionally, we have used red colour in the revised portions of the manuscript by using “Track Changes” function.

Hopefully the revised version could be acceptable, and if there are any improper revision, please don’t hesitate to inform us.

Great thanks again,

Qiyu Diao

Key Laboratory of Feed Biotechnology of Ministry of Agriculture and Rural Affairs, Feed Research In-stitute of Chinese Academy of Agricultural Sciences, Beijing, China

Response to Reviewer  Comments

Point 1: Line 71 - Add information on the origin of the Hu sheep and the feeds they were fed before your experiment started.

Response 1: Yes, accepted and revised It has been added.

The experimental Hu sheep were purchased in Huzhou, China. This study was carried out at the Chinese Academy of Agricultural Sciences' Nankou Pilot Base in Beijing, China (40° 21′ 19.7′′N, 116° 45′ 53.3′′E; 49–74 m height) from September to November 2020, lasting for 70 days. The pretest period lasted for 10 days and the formal period lasted for 60 days. The area has a flat topography and four distinct seasons. In order to accurately perceive the influence of day and night, the experimental sheep were fed outdoors in a single pen on a concrete floor, which prevented the Hu sheep from eating anything other than diet, thus making the experiment more accurate.

  Forty-five healthy male Hu sheep (weight 21.57 ± 0.77 kg, 2 months of age) were randomly divided into three treatment groups of 15 sheep each. They were fed the same concentrate and roughage in the ratio of 56:44 (see Table 1 for nutrition level). The sheep were fed twice daily following three different treatments or feeding modes: the control group (CON) Hu sheep were fed 50% of the total concentrate and roughage at sunrise, and the remaining 50% at sunset; the DH group was fed 70% of the total concentrate and 30% of the total roughage at sunrise and the rest at sunset; and the DL group was fed 30% of the total concentrate and 70% of the total roughage at sunrise and the rest at sunset. They were provided with clean fresh water. The study duration was two months, as shown in Table 2. They were kept outdoors to study the influence of the natural diurnal cycle of day and night more effectively.

Point 2: Line 76 - include approximate types of sunrise and sunset and the time of year the experiment was conducted.

Response 2: Yes, accepted and revised, It has been added.

Point 3: Line 76 - State exactly what the concentrate feed was. If it was a commercial product, state its name and the main ingredients.  The ingredients in the concentrate and the method used to prepare the concentrate will have an effect on your results.

Response 3: Yes, accepted and revised. The addition has been made, as shown in Table 1 below.

Here I would like to explain our purpose of this experiment was to investigate the effects of circadian rhythm on rumen environment, enzyme activity and microflora of Hu sheep, and on this basis, to explore whether it is better to feed with a high proportion of concentrate during the day or at night, so as to achieve precise feeding and reduce dietary waste. Since all the test sheep were fed concentrate from the same manufacturer, I don't think the preparation method of concentrate has a major impact on the test results. Thanks. 

Point 4: Line 80 - Describe the environment the sheep were housed in during your experiment.  Things the sheep could eat in the environment, such as dirt, could have an effect on rumen microorganisms.

Response 4: Yes, accepted and revised ,This part has been briefly described. Thanks .

 Point 5: Table 1 - Describe the wild rye. This is a feed this reviewer is not familiar with.  Please describe it.  Is it a dried straw or hay?

Response 5: Yes, accepted and revised .changed “ wild rye” to “Leymus chinensis” and added remarks below Table 1.

Point 6: Line 87 - Add manufacturer of the rumen catheter.

Response 6: Yes, accepted and revised, It has been added. Thanks. 

Point 7: Line 90 - Add manufacturer and model number of the pH meter.

Response 7: Yes, accepted and revised .It has been added.

Round 2

Reviewer 2 Report

Accept this paper

Author Response

Thank you so much.